# An Adaptive Signal Control Method with Optimal Detector Locations

**Senlai Zhu [1],\*, Ke Guo [2],\*, Yuntao Guo [3], Huairen Tao [1] and Quan Shi [1]**

[1]  School of Transportation, Nantong University, Nantong 226019, China; taohuairen@ntu.edu.cn (H.T.); shi.q@ntu.edu.cn (Q.S.)
[2]  JSTI GROUP, Nanjing 210019, China
[3]  Lyles School of Civil Engineering, Purdue University, West Lafayette, IN 47907, USA; guo187@purdue.edu
\*  Correspondence: zhusenlai@ntu.edu.cn (S.Z.); guoke0310@163.com (K.G.)

**Abstract:** The adaptive traffic signal control system is a key component of intelligent transportation systems and has a primary role in effectively reducing traffic congestion. The high costs of implementation and maintenance limit the applicability of the adaptive traffic signal control system, especially in developing countries. This paper proposes a low-cost adaptive signal control method that is easy to implement. Two detectors are installed in each vehicle lane at an optimal location determined by the proposed method to detect green and red redundancy time, based on which the original signal timing is adjusted through a signal controller. The proposed method is evaluated through case studies with low and high volume-to-capacity ratio intersections. The results show that the proposed adaptive signal control method can significantly reduce total traffic delay at intersections.

**Keywords:** adaptive signal control; detector location; signal timing; redundancy time; queue length; intelligent transportation systems

## 1. Introduction

The intersection signal control system plays a crucial role in intelligent transportation systems (ITS) to alleviate traffic congestion. The control scheme can be classified into non-adaptive [1–3] and adaptive control [4–6] methods. The major difference between these two methods is whether signal parameters can be adjusted in real-time with regard to detected traffic conditions.

In the non-adaptive traffic signal control method, a day is segmented into several time intervals (typically 3–5), and a signal timing plan is predetermined for each time interval. The basic premise is that the traffic pattern within each interval is relatively consistent and the predetermined signal timing is best suited for the condition at that time of day. The timing plan is often obtained using Webster's formula [7] or optimization tools such as TRANSYT [8] or TRANSYT-7F [9] with traffic flows (average flows, or highest flows) as inputs. Since traffic flows at intersections may vary significantly, an issue that traffic engineers may be confronted with is to determine what flows to use to optimize signal timings. Smith et al. [10] suggested using 90th percentile volumes as the representative volumes to generate optimal timing plans. Further, Yin [11] proposed three different models to determine robust optimal timing plans for isolated fixed-time signalized intersections and what flows to use for signal optimization. Recently, Smith [12] used a dynamical assignment and control model to design fixed-time or time of day signal timings for maximizing network capacity, and Marcianò et al. [13] studied the effects of signal setting configuration on user path choice behavior.

For adaptive traffic signal control methods, traffic signal timing parameters such as cycle length, phase split and duration of each phase adapt based on detected traffic conditions and traffic fluctuations

in order to achieve a set of specific objectives (such as minimizing the total traffic delay). In the 1980s, SCOOT and SCATS began pioneering the development of adaptive signal control system [14,15]. Then, along with the development of detection technologies, some adaptive signal control systems were proposed and even deployed in the field, e.g., RHODES [16] and TUC [17]. However, most adaptive signal control systems work with stage-based control in which phase sequence is predetermined. To generate phase sequences dynamically, some group-based control methods [18–21] have been proposed. Recently, some innovative methods such as machine learning, artificial intelligence and reinforcement learning have been used in the adaptive signal control method [22–27], and Jing et al. [28] proposed an adaptive signal control method in a connected vehicle environment.

Although the adaptive traffic signal control methods are more efficient in alleviating traffic congestion than the non-adaptive ones, the wide-scale implementation of such systems needs a significant amount of long-term investment, especially in developing countries, due to high implementation and maintenance costs. One of the main reasons that classical adaptive traffic signal control methods (including big data and transport models approaches) have high implementation and maintenance costs is these methods normally require various types of inputs such as detected and forecasted traffic volume of each direction, and queen length, etc. [16,29,30]. High-precision detectors/technologies (such as cameras, GPS) are needed to acquire these inputs with additional procedures to process them, hence the implementation and maintenance costs are high. There is a critical need to develop a low-cost adaptive traffic signal control method using inputs which can be detected easily.

In the literature, adaptive traffic signal control methods are normally developed for oversaturated intersections or intersections during oversaturated periods [31,32]. However, during off-peak periods, drivers may also experience delay, especially when there is no vehicle in directions that normally cause conflict points with the current direction. Hence, it is also worth studying adaptive traffic signal control methods for intersections during off-peak periods.

In addition, for both non-adaptive and adaptive methods, on-street detectors (such as in-pavement loop detectors) are deployed for the purpose of sensing intersection traffic conditions. The locations of detectors can significantly affect the accuracy of the detected traffic conditions. However, in the literature, few researches have tried to optimize the location of detectors.

To address the aforementioned gaps, this paper proposes an adaptive traffic signal control method to minimize the intersection traffic delay using the detected green and red redundancy time of each direction. The proposed method has a relatively low implementation cost and is easy to deploy in the field compared to classical adaptive traffic signal control methods as only two detectors such as widely used loop detectors are needed for each vehicle lane to provide sufficient data for traffic signal control. The proposed method is especially applicable for developing countries and rural areas, where traffic management budget is limited. In addition, methods used to determine the optimal location of each detector is presented for intersections with both low and high volume-to-capacity ratios.

In summary, the contributions of this paper are:

(1)  An adaptive traffic signal control method with low implementation and maintenance costs is proposed.
(2)  The locations of detectors are optimized for intersections with both low and high volume-to-capacity ratios.

The remainder of the paper is organized as follows. In Section 2, the adaptive traffic signal control method is proposed. In Section 3, the optimal location of each detector is discussed for intersections with both low and high volume-to-capacity ratios. Section 4 proposes case studies to verify the applicability of the proposed method and the location of detectors. Finally, some conclusions and discussions are provided in the last section.

## 2. Adaptive Signal Timing Control Method

In this section, to explain the proposed adaptive signal control method, we firstly introduce how to determine the number of detectors needed for each vehicle lane and what data should be collected by each detector. Then, the method to use the collected data for determining adaptive signal timing is discussed.

### 2.1. Data Collected by Detectors

In some intersections, especially those with low traffic volume or those during off-peak periods, the predeterimed signal timings of some phases sometimes are too large and have redundancy time, including green and red redundancy time when the traffic volume becomes small. Thus, the operation efficiency of the intersection is reduced. The key data of the proposed method is to collecte the green and red redundancy time of each phase. Adjusting the signal timing of each phase to reduce the total green and red redundancy time can ensure more timely transformation between signal phases and improve the operation level of the intersection.

To collect the green and red redundancy time, Figure 1 shows the relative locations of detectors (such as pavement inductive loop detectors connected with traffic signal) installed on the lane (including left turn, right turn and go straight). Note that for each vehicle lane, there are two detectors, denoted as detector A and detector B, among which the former is closer to the stop line. The distance between detector A and stop line is $D_A$ and the distance between detector B and stop line is $D_B$.

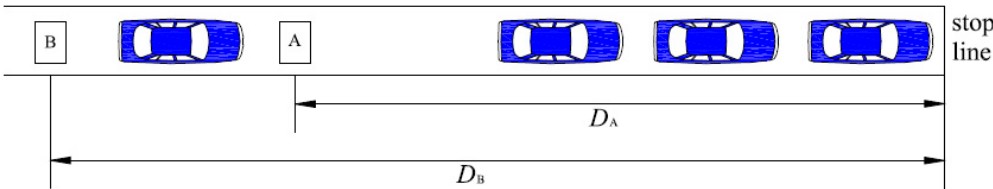

**Figure 1.** Location of detectors on the vehicle lane.

Figure 2 shows the adaptive traffic signal control system and its components, including data storage unit, signal controller, detectors on the vehicle lanes, and the signal timing of the intersection. The data storage unit (such as ROM/RAM) is to store the initial signal timing recorded by the controller (arrow 2), data collected by detectors (arrow 1), software for signal controller, operation system etc. The data will be used by the signal controller (arrow 3) to update signal timings using the proposed method (arrow 4), which is described in the next subsection.

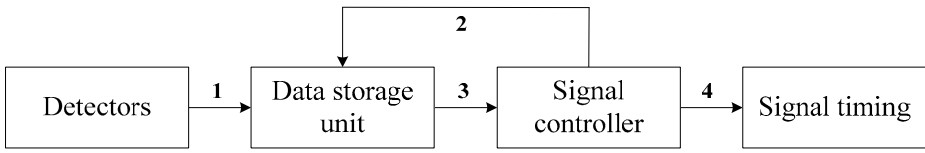

**Figure 2.** The adaptive traffic signal control system.

Given an initial intersection signal timing for a signal cycle with a green time $G_i$ and a red time $R_i$ for direction $i$ (recorded by the controller), when the corresponding green time for direction $i$ starts, the controller records the green light start time $TG_i$, detector A collects the time $LG_i$ of the last vehicle passing detector A during the green time. When the corresponding red time for direction $i$ starts, the controller records the red light start time $TR_i$, detector B collects the time $LR_i$ of the last vehicle passing detector B during the red time.

In summary, the initial green time $G_i$, red time $R_i$, the green and red light start time ($TG_i$ and $TR_i$) of each phase are stored in the data storage unit. The time $LG_i$ of the last vehicle passing detector A during the green time of each phase is collected by detector A, and the time $LR_i$ of the last vehicle

passing detector B during the red time of each phase is collected by detector B. The green redundancy time $GR_i$ for direction $i$ during a specific signal cycle can be determined by $GR_i = G_i - LG_i + TG_i$, and the red redundancy time $RR_i$ for direction $i$ can be determined by $RR_i = R_i - LR_i + TR_i$.

### 2.2. Method to Adapt the Signal Timing

This subsection illustrates the method to adapt the signal timing (arrow 4 in Figure 2). When the green time or red time of some phases are set too large and/or the traffic volume is relatively small, the total operational efficiency of the intersection is reduced. For a direction, once the redundancy time $GR_i$ and $RR_i$ are detected and computed, the green time $\overline{G}_i$ and red time $\overline{R}_i$ for the next phase cycle can be adjusted as $\overline{G}_i = G_i - GR_i$ and $\overline{R}_i = R_i - RR_i$. However, for a cycle of one intersection, there are several signal phases and each phase consists of several directions. To optimize the signal timing of the intersection, we cannot adapt each direction separately.

Denote $\mathbf{P} = [1, 2, 3, \ldots, m, \ldots]$ as the set of phases in one signal cycle, $G_m$ as the green time of phase $m \in \mathbf{P}$, $R_m$ as the red time of phase $m$, $GR_m^{min}$, $RR_m^{min}$ as the smallest green and red redundancy time of all directions belong to phase $m$, respectively. The procedure to adapt the signal timing of the intersection is shown in Figure 3. Specifically, each step of the procedure is given as follows:

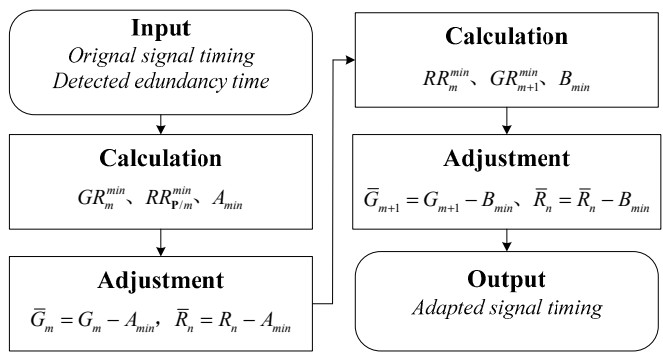

**Figure 3.** Procedure to adapt signal timing.

**Step 1**: Take arbitrary phase $m \in \mathbf{P}$ as the basic adjustment phase (without loss of generality, we can choose the first phase). Calculate the smallest green redundancy time $GR_m^{min}$ of all directions belonging to this phase, and calculate the smallest red redundancy time of all directions belonging to all phases, except $m$ (i.e., $\mathbf{P}/m$) denoted as $RR_{\mathbf{P}/m}^{min}$. Calculate $A_{min} = min\{GR_m^{min}, RR_{\mathbf{P}/m}^{min}\}$.

**Step 2**: For the next signal cycle, adjust the green time of phase $m$ by $\overline{G}_m = G_m - A_{min}$, and adjust the red time of phase $n \in \mathbf{P}/m$ by $\overline{R}_n = R_n - A_{min}$.

**Step 3**: For phase $m$ (same with the chosen phase in Step 1), calculate the smallest red redundancy time $RR_m^{min}$ of all directions belong to this phase. Calculate the smallest green redundancy time $GR_{m+1}^{min}$ of all directions belong to the next phase $m + 1$. Calculate $D_{min} = min\{RR_n^{min} - A_{min}, \forall n \in \mathbf{P}/\{m, m+1\}\}$. Calculate $B_{min} = min\{RR_m^{min}, GR_{m+1}^{min}, D_{min}\}$.

**Step 4**: For the next signal cycle, adjust the green time of phase $m + 1$ by $\overline{G}_{m+1} = G_{m+1} - B_{min}$ and adjust the red time of phase $n \in \mathbf{P}/m + 1$ by $\overline{R}_n = \overline{R}_n - B_{min}$ where $\overline{R}_n$ in the right side of the equation is the updated value in Step 2. This is the end of the procedure.

Note that the proposed method is to adapt the signal timing based on the detected green and red redundancy time of each phase; the procedure is easy to process. Since only two detectors (inductive loop detectors) are needed for each vehicle lane, the proposed method is low-cost and easy to implement. The adapted signal timing is based on the detected conditions of the previous signal cycle and does not consider traffic forecasts. This is reasonable, since the traffic pattern within a certain time period of a day is relatively consistent (similar to the basic premise of the non-adaptive traffic signal control methods), and the condition of the previous signal cycle is normally more reliable than traffic forecasts, which tend to have high forecast errors.

## 3. Location of Detectors

The key of the proposed adaptive signal timing method is to detect and compute the redundancy time $GR_i$ and $RR_i$ of each direction, and the accuracy of the detection is closely related to the location of each detector (i.e., $D_A$ and $D_B$ in Figure 1). This section shows how to optimize the location of each detector.

The proposed adaptive signal method tries to reduce the green and red redundancy time of each signal phase, thus aiming to reduce the total vehicle delay for an intersection. The expected queue length is normally adopted to quantify vehicle delay for an intersection and reflect the operational level of the considered intersection. Hence, to capture the traffic delay of each direction actually, we first explain how to estimate the expected queue length, and then determine optimal detector locations.

Without loss of generality, the method based on probe vehicles proposed by Comert [33] to estimate the expected queue length is adopted. Since different intersections with different traffic volumes have different characteristics, two kinds of intersections are considered—intersections with low and high volume-to-capacity ratios (denoted as ρ). As proofed by Comert [33], when ρ > 80%, the overflow queue reveals a significant impact on the estimation errors. Thus, 80% in this paper is chosen as the volume-to-capacity ratio threshold value between the two kinds of intersections.

In this paper, the waiting lanes are assumed to have infinite capacity, the vehicle is assumed to accelerate and decelerate instantaneously, the arrival of vehicles is assumed to follow Poisson distribution, and the vehicle accumulation is assumed to be a vertical queue.

For direction $i$, let $S_i$ be the last probe vehicle location and $N_i$ be the queue length. When ρ ≤ 80%, the expected last probe vehicle location is determined by:

$$E(S_i) = \lambda_i R_i - \frac{(1 - p_i)(1 - e^{-p_i \lambda_i R_i})}{p_i} \tag{1}$$

where for traffic direction $i$, $E(S_i)$ is the expected last probe vehicle location; $p_i$ is the rate of probe vehicle; $\lambda_i$ is the arrival rate; $R_i$ is the red time.

Note that $E(S_i)$ gets closer to the queue length as the probe $p_i$ rate increases, thus the expected queue length $E(N_i)$ of traffic direction $i$ can be obtained by:

$$E(N_i) = \lim_{p_i \to 1} \lambda_i R_i - \frac{(1 - p_i)(1 - e^{-p_i \lambda_i R_i})}{p_i} = \lambda_i R_i \tag{2}$$

When ρ > 80%, the queue length will exceed $\lambda_i R_i$ with the increase of traffic flow because an overflow queue (the leftover queue from a previous cycle) may exist. For direction $i$, let $Q_i$ be the overflow queue length and $A_i$ be the queue length that occurs due to new arrivals during the red duration. The total queue length $N$ is written as the summation of $Q_i$ and $A_i$. Depending on the position of last probe vehicle, three possible scenarios can be obtained: (1) The last probe may be within the overflow queue ($s_i \in Q_i$); (2) The last probe may be present within new arrivals ($s_i \in A_i$); (3) There may not be any probe vehicle in the queue at all ($s_i = 0$).

Then, for direction $i$, given the rate of probe vehicle $p_i$, the expected queue length can be obtained by:

$$\begin{aligned} E(N_i|p_i) = P(s_i \in Q_i|p_i)[E(Q_i) + \theta_i R_i] + P(s_i \in A_i|p_i)[E(Q_i) + \lambda_i R_i] \\ + P(s_i = 0|p_i)[(1 - p_i)(E(Q_i) + \theta_i R_i)] \end{aligned} \tag{3}$$

where $\theta_i = (1 - p_i)\lambda_i$.

In Equation (3), the expected value of the overflow queue $E(Q_i)$ is used from Akçelik [34]:

$$E(Q_i) = 1.5\left(\rho_i - \rho_i^0\right)/(1 - \rho_i) \tag{4}$$

where for direction $i$, $\rho_i = \frac{\lambda_i C}{d_i G_i}$, $\rho_i^0 = 0.67 + \frac{d_i G_i}{600}$, $C$ is the cycle time, $d_i$ is departure rate, other parameters remain the same as described above.

Probabilities $P(s_i \in Q_i | p_i)$, $P(s_i \in A_i | p_i)$ and $P(s_i = 0 | p_i)$ in Equation (3) are approximately by [34]:

$$P(s_i = s | p_i) = \begin{cases} \left[1 - e^{-p_i[(9.87p_i{}^2 - 4.62p_i + 0.991)E(Q_i)]}\right] e^{-p_i \lambda_i R_i} & s_i \in Q_i \\ e^{-p_i \lambda_i R_i} & s_i \in A_i \\ e^{-p_i[(9.87p_i{}^2 - 4.62p_i + 0.991)E(Q_i)]} & s_i = 0 \end{cases} \tag{5}$$

Similarly, $E(N_i | p_i)$ gets closer to the queue length as the probe $p_i$ rate increases; thus, the expected queue length $E(N_i)$ of traffic direction $i$ can be obtained by:

$$E(N_i) = \lim_{p_i \to 1} E(N_i | p_i) = P(s_i \in Q_i) E(Q_i) + P(s_i \in A_i)[E(Q_i) + \lambda_i R_i] \tag{6}$$

where $E(Q_i)$ is obtained by Equation (4), probabilities $P(s_i \in Q_i)$ and $P(s_i \in A_i)$ are approximately by:

$$P(s_i = s) = \begin{cases} \left[1 - e^{-6.242 E(Q_i)}\right] e^{-\lambda_i R_i} & s_i \in Q_i \\ e^{-\lambda_i R_i} & s_i \in A_i \end{cases} \tag{7}$$

In summary, when $\rho \le 80\%$, the expected queue length can be obtained by Equation (2), and when $\rho > 80\%$, the expected queue length can be obtained by Equations (4), (6) and (7).

Once the expected queue length is determined for each direction, the location of detector A and B are determined by:

$$D_A = |E(N_i)| \times l_b + (|E(N_i)| - 1) \times l_a \tag{8}$$

$$D_B = V_{s,i} \times G_i \tag{9}$$

where $|E(N_i)|$ is the integer number of $E(N_i)$, $l_b$ is the length of a standard vehicle; $l_a$ is the average head spacing, $V_{s,i}$ is the designed maximum speed of the vehicle lane for direction $i$.

As described above, detector A detects the green light start time $TG_i$ and the time $LG_i$ of the last vehicle passing detector A during the green time. And detector B detects the red light start time $TR_i$ and the time $LR_i$ of the last vehicle passing detector B during the red time. That is, detector A is to detect and compute the green redundancy time $GR_i$ and detector B is to detect and compute the red redundancy time $RR_i$. Thus, detector A should be located around the expected position of the last vehicle in the queue from the stop line, and detector B should be located around the maximum distance from the stop line that a vehicle can travel during the green time.

Note that for some intersections, the volume-to-capacity ratio may be low during some time periods (i.e., peak periods) and may be high during other time periods (i.e., off-peak periods). In this case, if we want to keep the adaptive traffic signal control system running all the time, we can set three detectors for each vehicle lane (detector A, B, C). The location $D_B$ of detector B is determined by Equation (9), the location $D_A$ of detector A is determined by Equation (8), the expected vehicle queue length $E(N_i)$ is determined by Equation (2), location $D_c$ of detector C is determined by Equation (8), and the expected vehicle queue length $E(N_i)$ is determined by Equations (4), (6) and (7). Thus, the adaptive traffic signal control system operates based on traffic conditions detected by detector A, B during off-peak periods, and on traffic conditions detected by detector C, B during peak periods.

## 4. Case Studies

To verify the effectiveness of the proposed adaptive signal control method, we analyzed the proposed method through case studies in two intersections of Suzhou, China, since locations of detectors are different for intersections with different volume-to-capacity ratios (i.e., $\rho$). We first tested the proposed method on the Linquan-Wenjing intersection with a low volume-to-capacity ratio,

ρ ≤ 80% (denoted as intersection 1 in Figure 4), and then on the Xinghu-Xiandai intersection with a high volume-to-capacity ratio, ρ > 80% (denoted as intersection 2 in Figure 4).

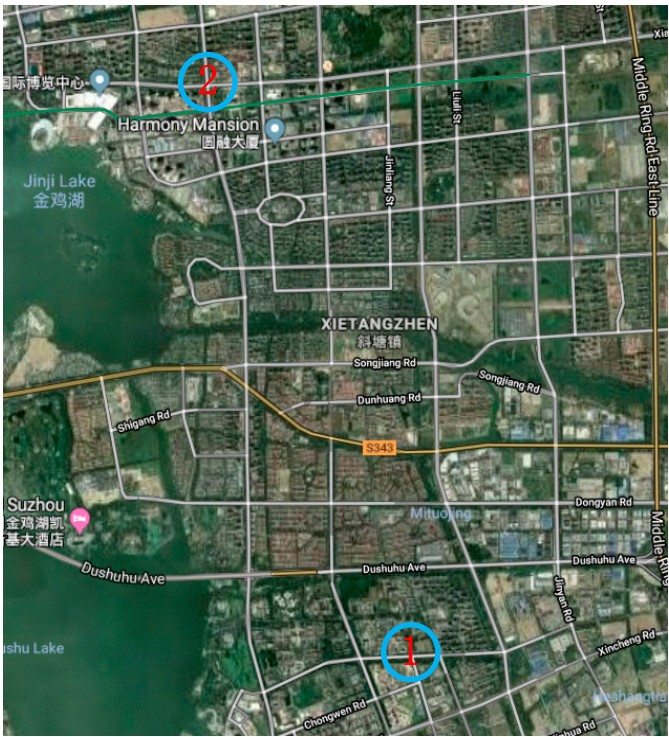

**Figure 4.** Test intersections of Suzhou, China.

For the Linquan-Wenjing intersection, Figure 5 shows the original signal timing. As shown in the figure, each signal cycle has four phases: (1) north and south straight, (2) north and south left, (3) east and west straight, and (4) east and west left. The corresponding green time is set to be 20, 20, 25 and 19, respectively (the unit is second, which is omitted for illustration purposes, and the same below), the yellow time are set to be 3 for all phases, and the red time are 73, 73, 68, and 74, respectively. The traffic volume and travel speed of five signal cycles for each direction are collected by field studies and the traffic volume is treated as fixed traffic flow input for our simulation. Table 1 shows the initial traffic volume of the Linquan-Wenjing intersection in vehicle through per signal cycle length and the initial signal cycle length is 96 s. For example, in the third signal cycle, traffic flow of north straight is 5, which means that 5 vehicles travelled north straight during the third signal cycle. Note that the signal cycle length will change once the proposed adaptive signal control method is implemented.

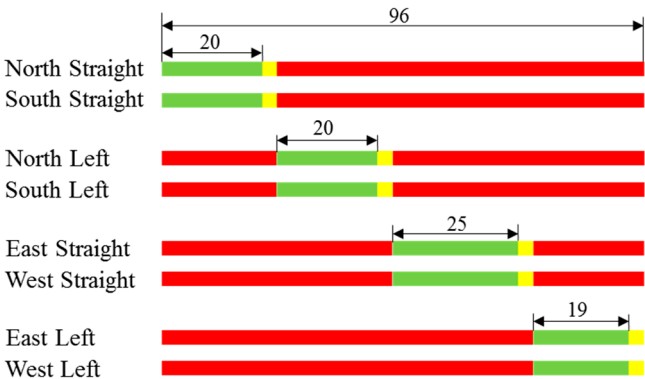

**Figure 5.** Original signal timing of the Linquan-Wenjing intersection.

**Table 1.** Traffic volume (vehicles/signal cycle) of Linquan-Wenjing intersection.

| Signal Cycle | Travel Direction | | | | | | | |
|---|---|---|---|---|---|---|---|---|
| | North Straight | North Left | South Straight | South Left | East Straight | East Left | West Straight | West Left |
| 1 | 3 | 3 | 4 | 0 | 3 | 2 | 3 | 2 |
| 2 | 0 | 0 | 6 | 2 | 1 | 5 | 1 | 3 |
| 3 | 5 | 1 | 4 | 1 | 0 | 1 | 2 | 2 |
| 4 | 1 | 1 | 3 | 2 | 0 | 0 | 1 | 0 |
| 5 | 5 | 5 | 2 | 3 | 2 | 0 | 0 | 0 |

Figure 6 shows the computed redundancy time for each cycle by the proposed method with the adapted signal timing for each signal cycle. Take the first signal cycle for example; it can be seen from Figure 6 that the computed green and red redundancy time ($GR_m^{min}$, $RR_m^{min}$) for each phase is (1, 2), (2, 2), (7, 8) and (15, 17), respectively. Thus, using the procedure of the proposed adaptive signal timing control method introduced in Section 2:

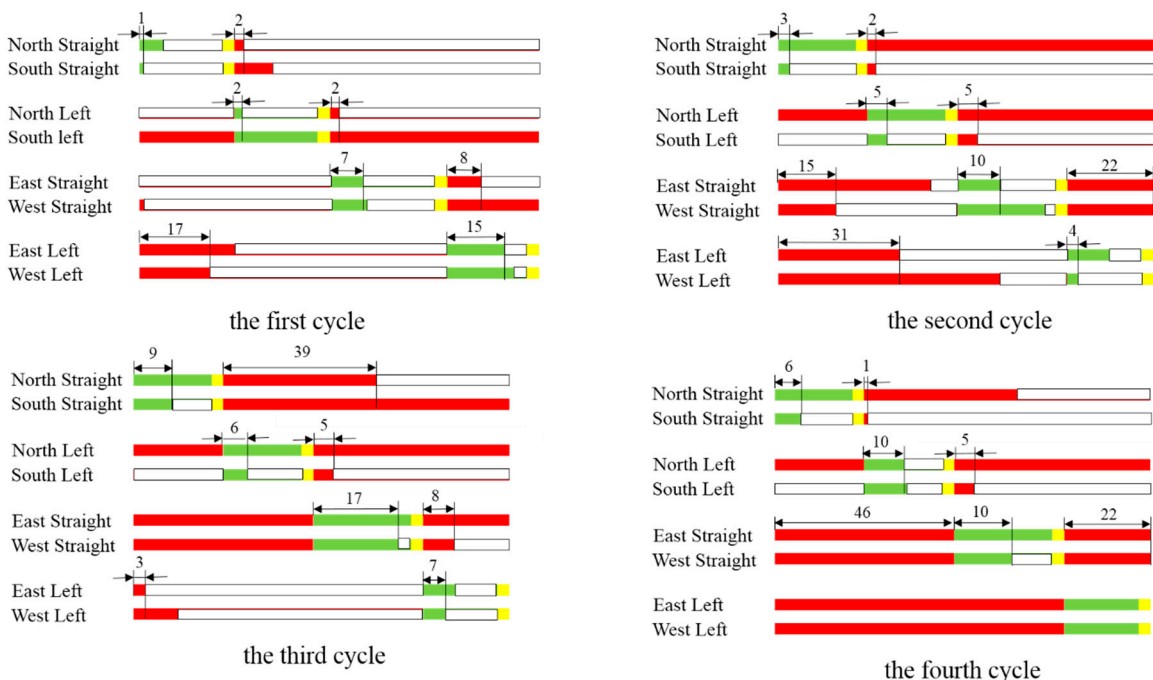

**Figure 6.** Redundancy time.

**Step 1**: We take the first phase as the basic adjustment phase. The smallest green redundancy time $GR_1^{min}$ of all directions belonging to this phase is 1; and the smallest red redundancy time $RR_{\mathbf{P}/1}^{min}$ of all directions belonging to phases except the first phase is 2. $A_{min} = min\{GR_1^{min}, RR_{\mathbf{P}/1}^{min}\} = 1$.

**Step 2**: For the next signal cycle, adjust the green time of the first phase by $\overline{G}_1 = G_1 - A_{min} = 20 - 1 = 19$, and adjust the red time of phase $n \in \mathbf{P}/1$ by $\overline{R}_n = R_n - A_{min}$, i.e., $\overline{R}_2 = R_2 - A_{min} = 73 - 1 = 72$, $\overline{R}_3 = R_3 - A_{min} = 68 - 1 = 67$, $\overline{R}_4 = R_2 - A_{min} = 74 - 1 = 73$.

**Step 3**: The smallest red redundancy time $RR_1^{min}$ of all directions is 2. The smallest green redundancy time $GR_2^{min}$ of all directions belonging to the second phase is 2. $D_{min} = min\{RR_n^{min} - A_{min}, \forall n \in \mathbf{P}/\{1, 2\}\} = 7$. $B_{min} = min\{RR_1^{min}, GR_2^{min}, D_{min}\} = 2$.

**Step 4**: For the next signal cycle, adjust the green time of the second phase by $\overline{G}_2 = G_2 - B_{min} = 20 - 2 = 18$ and adjust the red time of phase $n \in \mathbf{P}/2$ by $\overline{R}_n = R_n - B_{min}$, i.e., $\overline{R}_1 = R_1 - B_{min} = 73 - 2 = 71$, $\overline{R}_3 = R_3 - B_{min} = 67 - 2 = 65$, $\overline{R}_4 = R_4 - B_{min} = 73 - 2 = 71$.

After this, the adapted signal timing of the next signal cycle is determined, as shown in Figure 7. Similarly, the timing of the next several signal cycles can be determined by the proposed method.

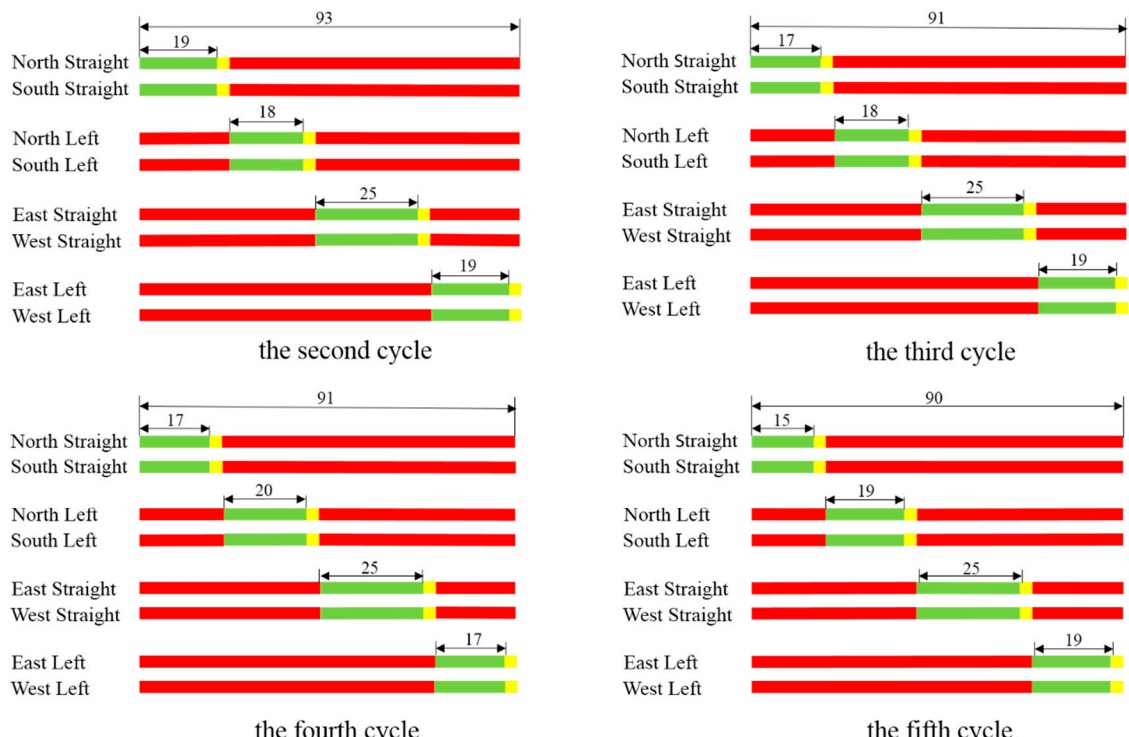

**Figure 7.** Adapted signal timing.

Further, to verify the effectiveness of the proposed method and the optimized signal timing, we compare the adapted signal timing (shown in Figure 7) with the original signal timing (shown in Figure 5) in VISSIM simulations. Figure 8 shows the average delay of the intersection under different signal timing patterns. In Figure 8, "before" denotes the original signal timing, and "after" denotes the adapted signal timing. Results show that the adapted signal timing can reduce the total average delay by about 10% for each signal cycle. This verifies the effectiveness of the proposed method on intersection with low traffic volume.

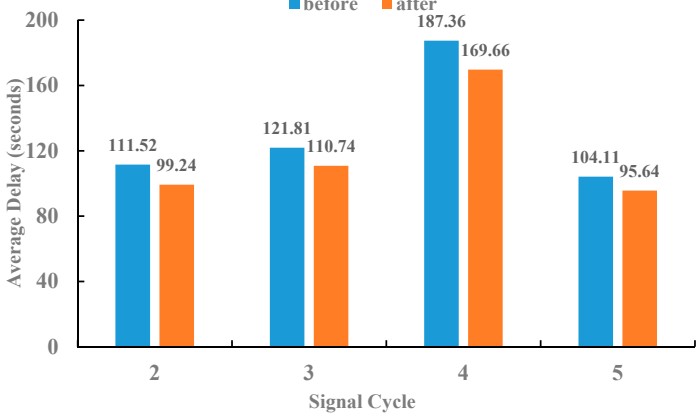

**Figure 8.** Average delay of the Linquan-Wenjing intersection.

Similar to the simulation on the Linquan-Wenjing intersection, we test our method on the Xinghu-Xiandai intersection with ρ > 80%. Similarly, we investigate the original signal timing and detect the traffic volume and travel speed of five signal cycles for each direction artificially. Figure 9 compares the updated signal timing with the original signal timing in VISSIM simulations. It can be seen that the delay of each cycle under the adapted signal timing is smaller than that under the original signal timing.

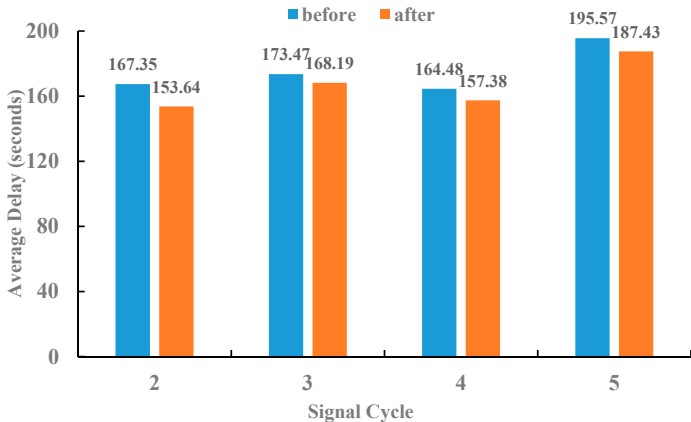

**Figure 9.** Average delay of the Xinghu-Xiandai intersection.

Matched-pairs t-tests are preformed to test whether there is a significant mean difference between the total delays of each signal cycle before and after introducing the proposed method at both intersections. The results show that the total delays of each signal cycle of Linquan-Wenjing and Xinghu-Xiandai intersections are statistically significant ($p = 0.004$ and $p = 0.009$, respectively). It illustrates that the proposed method can lead to a statistically significant reduction in total delays at both intersections.

Table 2 shows the delay reduction of Linquan-Wenjing and Xinghu-Xiandai intersections using the proposed adaptive signal control method. It can be seen that the adapted signal timing can reduce the total average delay of Linquan-Wenjing and Xinghu-Xiandai intersections by about 10% and 4%, respectively, for one signal cycle. It illustrates that the proposed method is more effective in reducing the traffic delay of intersection with low volume-to-capacity ratios compared to that of intersection with high volume-to-capacity ratios. This is because for intersections with high volume-to-capacity ratios, the detected green and red redundancy time are normally very small and even equal to zero; thus the adapted signal timing is very close to the original one. Hence, the proposed method is more applicable to intersections with low volume-to-capacity ratios or those during off-peak periods.

**Table 2.** Traffic delay (seconds) reduction of Linquan-Wenjing and Xinghu-Xiandai intersections.

| Intersection | | Total Delay of Each Signal Cycle | | | |
|---|---|---|---|---|---|
| | | **2** | **3** | **4** | **5** |
| Linquan-Wenjing | Before introducing the proposed method | 111.52 | 121.81 | 187.36 | 104.11 |
| | After introducing the proposed method | 99.24 | 110.74 | 169.66 | 95.64 |
| | Reduction rate | 11.01% | 9.09% | 9.45% | 8.14% |
| Xinghu-Xiandai | Before introducing the proposed method | 167.35 | 173.47 | 164.48 | 195.57 |
| | After introducing the proposed method | 153.64 | 168.19 | 157.38 | 187.43 |
| | Reduction rate | 8.19% | 3.04% | 4.32% | 4.16% |

Finally, to verify the detector locations determined by the proposed method, we simulate several different detector locations. Five cases are considered for both Linquan-Wenjing and Xinghu-Xiandai intersections, as shown in Figure 10.

Note that in Case 1, detector locations are determined by the proposed method (Equations (8) and (9)), that is $\overline{D}_A = D_A$ and $\overline{D}_B = D_B$. Case 2 and 3 set detector B with the same location and proposed method, and set detector A with the location closer and farther from the stop line than that of the proposed method, respectively (i.e., Case 2: $\overline{D}_A = 0.5 \times D_A, \overline{D}_B = D_B$; Case 3: $\overline{D}_A = 1.5 \times D_A$, $\overline{D}_B = D_B$). Case 4 and 5 set detector A with the location same and with the proposed method, respectively, and set detector B with the location closer and farther from the stop line than that of

the proposed method, respectively (i.e., Case 4: $\overline{D}_A = D_A$, $\overline{D}_B = 0.5 \times D_B$; Case 5: $\overline{D}_A = D_A$, $\overline{D}_B = 1.5 \times D_B$).

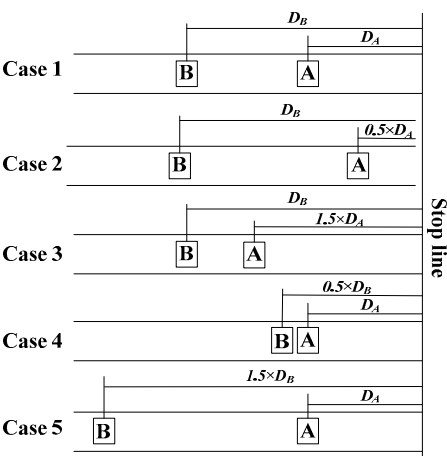

**Figure 10.** Test cases with different detector locations.

Table 3 compares the traffic delay under the adaptive signal timing derived from different detector locations (i.e., Case 1–5) for both Linquan-Wenjing and Xinghu-Xiandai intersections. It can be seen that the traffic delay from detector locations determined using the proposed method in Case 1 is the smallest. This is because that if detector A is located too close to the stop line, the green redundancy time may be underestimated (may always be zero for all directions), and if detector A is located too far from the stop line, the green redundancy time may be overestimated. Similarly, the red redundancy time may be underestimated or overestimated if detector B is located too close or too far. This verifies the reasonability of the detector locations determined by the proposed method.

**Table 3.** Traffic delay (seconds) under the adapted signal timing from different detector locations

| Intersection | Detector Location | | | | |
|---|---|---|---|---|---|
| | Case 1 | Case 2 | Case 3 | Case 4 | Case 5 |
| Linquan-Wenjing | 108 | 134 | 146 | 137 | 152 |
| Xinghu-Xiandai | 169 | 183 | 203 | 192 | 217 |

## 5. Conclusions and Discussions

To reduce the total traffic delay at intersections, this paper proposes a low-cost adaptive signal control method that can be easily implemented in the real world. In the proposed method, two detectors are installed for each vehicle lane to detect and compute the green and red redundancy time, respectively. Based on the detected redundancy time, the original signal timing is adjusted.

Further, the optimal location of each detector is discussed for intersections with low and high volume-to-capacity ratios. Detectors to sense the green redundancy time should be located around the expected position of the last vehicle in the queue from the stop line, and detectors to sense the red redundancy time should be located at about the maximum distance that a vehicle can travel during the green time before the stop line.

Case studies were conducted on two intersections with different volume-to-capacity ratios. The results show that the proposed adaptive signal control method can significantly reduce the total traffic delay of intersections with both low and high traffic volume. Through the comparison of cases with different sensor locations, it is found that under the location optimized by the proposed method, the total traffic delay based on the adaptive signal control method is the smallest.

The proposed signal control method in this study is developed for an intersection. A possible extension of this work is to factor signal coordination in a traffic network with multiple intersections.

Traffic forecasts were not considered in the proposed method and additional research is needed to factor traffic forecast in the adaptive signal control methods. In addition, the signal phase in the proposed signal control method remains unchanged. Another extension to this work is to study methods to adapt the signal phase along with signal timing based on detected traffic conditions.

**Author Contributions:** S.Z. and K.G. collected the data; S.Z. formulated the method and designed the procedure; K.G. analyzed the case studies; S.Z. approved the submission and publication; Q.S. supervised the project; S.Z., K.G., Y.G. and H.T. wrote the paper.

**Funding:** This research was funded by the Humanities and Social Science Foundation of the Ministry of education in China grant number [18YJCZH274].

**Acknowledgments:** This work is supported by the Humanities and Social Science Foundation of the Ministry of education in China (18YJCZH274). The authors are grateful for comments made by anonymous referees.

**Conflicts of Interest:** The authors declare no conflict of interest.

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
