# Peer review of "An Adaptive Signal Control Method with Optimal Detector Locations"

_sustainability, doi:10.3390/su11030727_

Reviewer 1 Report

The paper proposes a method to improve road network performances by means of an adaptive signal control of a road intersection. The attempts is to propose a “low-cost adaptive signal control method”.

The issue is relevant in the context of transport engineering and in particular in ITS applications. The issue is relevant in urban transport planning and in particular for cities located in developing countries.

The studied issue in the paper is relevant. Despite the relevance of the topic, the current version of the paper should be improved in order to improve its quality and readability. In the follow, there are suggestions, mayor and minor revisions.

Author Response

Point 1: The current version of the paper clarifies which are the contributions (lines 68-72). However, despite I agree to you that the adaptive traffic signal control methods require “high implementation and maintenance costs”, in my opinion you are proposing a simplified and practical procedure, inserting some little adjustments on signal regulation, rather than a classical transportation planning method. This could be a strength of your work but it is necessary to clarify differences between big data and transport models approaches.

 Response 1: Thank you for the comment. One of the main reasons that classical adaptive traffic signal control methods (including big data and transport models approaches) have high implementation and maintenance costs is these methods normally require various types of inputs such as detected and forecasted traffic volume of each direction, and queen length, etc. High-precision detectors/technologies (such as cameras, GPS) are needed to acquire these inputs with additional procedures to process them, hence the implementation and maintenance costs are high. To address this, the proposed method only requires the redundancy signal time from loop detectors which is easier and cheaper to collect than other traffic conditions, especially when the budget for traffic management is limited.

To clarify this, the revised paper includes the following statement in page 1 line 55:

“One of the main reasons that classical adaptive traffic signal control methods (including big data and transport models approaches) have high implementation and maintenance costs is these methods normally require various types of inputs such as detected and forecasted traffic volume of each direction, and queen length, etc. [29, 30]. High-precision detectors/technologies (such as cameras, GPS) are needed to acquire these inputs with additional procedures to process them, hence the implementation and maintenance costs are high. There is a critical need to develop a low-cost adaptive traffic signal control method using inputs which can be detected easily.

In addition, the following statements area also added in page 1 line 73:

“The proposed method has a relatively low implementation cost and easy to deploy in the field compared to classical adaptive traffic signal control methods as only two detectors such as widely used loop detectors are needed for each vehicle lane to provide sufficient data for traffic signal control. The proposed method is especially applicable for developing countries and rural areas where traffic management budget is limited.”

 Point 2: It is needed to highlight principal limits of your approach. Your practical procedure not consider traffic forecasts related to a network and it is limited to a partial portion of the system (e.g. more intersections). However, in some real context, where there are not any control strategy, your “low cost” procedure could find some usefulness. For this reasons, I suggest you to stress the simplifications introduced in your proposed approach, distinguishing it from the classic and expensive traffic signal control methods and ITS.

 Response 2: Thank you for the comment. To illustrate the applicability of the proposed approach, we clarify it in the conclusions section:

“The proposed signal control method in this study is developed for an intersection. A possible extension of this work is to factor signal coordination in a traffic network with multiple intersections.”

We also agree that the proposed method does not factor traffic forecasts at a network level and it is limited to a partial portion of the system. To clarify this, the revised paper includes the following statement in page 5 line 161:

“The adapted signal timing is based on the detected conditions of the previous signal cycle and does not consider traffic forecasts. This is reasonable since that the traffic pattern within certain time period of a day is relatively consistent (similar to the basic premise of the non-adaptive traffic signal control methods), and the condition of the previous signal cycle is normally more reliable than traffic forecasts which tend to have high forecast errors.”

 Point 3: Mayor revisions regard state of the art, the proposed method and some aspects of application. In relation to the state of the art, as reported among the Suggestions, it is needed to recall some scientific literature. In the Suggested references I report some indications.

 Response 3: Thank you for the comment. According to the suggestion, we have added some scientific literature.

 Point 4: In relation to the proposed method, despite the research contribution declared in the introduction and the intention to present a simplified approach, the current illustration of steps, its sequence and their mutual interaction is not fully clear. Maybe a figure representing a flow chart, indicating inputs, procedures and outputs could be useful in order to aid the reader to comprehend the whole procedure. The number of steps and the number in the figure 2, do have the same meaning?

 Response 4: Thank you for the comment. Figure 2 illustrates the components of the proposed adaptive signal control system, while the steps are used to describe the process to adapt the signal timing. Hence, the number of steps and the number in the figure 2 do not have the same meaning. To better illustrate the steps for adapting the signal timing, we added a figure [Figure 3] in the revised paper in page 4 line 138. And based on the figure, the description of these steps is further modified in the revised paper.

 Point 5: If I understand correctly, you adjust the duration of phases ex post. Then you simulate with VISSIM to reproduce the observed data. The procedure not implies a simulation for forecasting. Then, this is a simplified procedure. Is correct my interpretation?

According to my suggestions, I think that in the conclusions principal limits of your proposed procedure should be highlighted.

Response 5: Thank you for the comment. Your interpretation is correct. As also discussed in Point 2, the proposed method does not consider traffic forecasts. To clarify this, the revised paper includes the following statement in the conclusion section in page 12 line 340:

“Traffic forecasts were not considered in the proposed method and additional research is needed to factor traffic forecast in the adaptive signal control methods.”

 Point 6: In relation to the application, in table 1, the meaning of the numbers is not fully clear. These numbers are traffic flows in a cycle length? These numbers are traffic flows measured and fixed? Please, specify these points!

Please, indicate units of measure in the table 1.

Please, verify carefully the form of Table 1

 Response 6: Thank you for the comment. The numbers in table 1 are traffic flows in a signal cycle length. And these numbers are collected by field study and assumed to be fixed during the simulations. To clarify this, the revised paper includes the following statement in the revised paper in page 7 line 243:

“The traffic volume and travel speed of five signal cycles for each direction are collected by field studies and the traffic volume is treated as fixed traffic flow input for our simulation. For example, as shown in Table 1, in the first signal cycle, traffic flow of north straight is 3, which means that 3 vehicles related to the direction of north straight arrive during the first signal cycle.”

Reviewer 2 Report

The conducted research study is really interesting; unfortunately the paper has sever major drawbacks, requiring improvements:

- Cost is reported as one of main benefits. Therefore some price ranges should be listed to enable comparison.

- Differences cannot be estimated based only on visual comparison in Figs. 7 and 8; they need to be compared statistically.

- Discussion section is missing - this is where authors should list potential biases, limitations, comparison to previous studies, etc.

- Throughout the paper, terminology is sometimes confusing. For example, section "Case study" (i.e. singular) mentions "several intersections" (which sounds like many), while in fact, two case studies were used. Authors are also inconsistent in using both "adaptive signal method" and "adapt signal method". They also unexpectedly mix past and present tense.

- In addition, the text contains high number of typographical errors (queen instead of queue, volumn instead of volume) and strange expressions ("years away", "signal timing is wasted", "polished this paper").

The mentioned points should be improved, including corrections by a native speaker.

Author Response

Point 1: The conducted research study is really interesting; unfortunately the paper has sever major drawbacks, requiring improvements: Cost is reported as one of main benefits. Therefore some price ranges should be listed to enable comparison.

 Response 1: Thank you for the comment. One of the main reasons that classical adaptive traffic signal control methods (including big data and transport models approaches) have high implementation and maintenance costs is these methods normally require various types of inputs such as detected and forecasted traffic volume of each direction, and queen length, etc. High-precision detectors/technologies (such as cameras, GPS) are needed to acquire these inputs with additional procedures to process them, hence the implementation and maintenance costs are high. To address this, the proposed method only requires the redundancy signal time from loop detectors which is easier and cheaper to collect than other traffic conditions, especially when the budget for traffic management is limited.

Since different systems use different equipment and technologies, it is difficult to list the price ranges. And to clarify why cost is one of the main benefits of the proposed system, the revised paper includes the following statement in page 1 line 55:

“One of the main reasons that classical adaptive traffic signal control methods (including big data and transport models approaches) have high implementation and maintenance costs is these methods normally require various types of inputs such as detected and forecasted traffic volume of each direction, and queen length, etc. [29, 30]. High-precision detectors/technologies (such as cameras, GPS) are needed to acquire these inputs with additional procedures to process them, hence the implementation and maintenance costs are high. There is a critical need to develop a low-cost adaptive traffic signal control method using inputs which can be detected easily.

And in page 1 line 73:

“The proposed method has a relatively low implementation cost and easy to deploy in the field compared to classical adaptive traffic signal control methods as only two detectors such as widely used loop detectors are needed for each vehicle lane to provide sufficient data for traffic signal control. The proposed method is especially applicable for developing countries and rural areas where traffic management budget is limited.”

 Point 2: Differences cannot be estimated based only on visual comparison in Figs. 7 and 8; they need to be compared statistically.

 Response 2: Thank you for the comment. Taking the advice, we add a table [Table 2] in the revised paper to compare the differences in Figs. 7 and 8 statistically in page 11 line 304.  And we also add some discussions related to this table in page 10 line 294:

“Table 2 shows the delay reduction of Linquan-Wenjing and Xinghu-Xiandai intersections using the proposed adaptive signal control method. It can be seen that the adapted signal timing can reduce the total average delay of Linquan-Wenjing and Xinghu-Xiandai intersections by about 10% and 4% respectively for one signal cycle. It illustrates that the proposed method is more effective in reducing the traffic delay of intersection with low volume-to-capacity ratios compared to that of intersection with high volume-to-capacity ratios.”

 Point 3: Discussion section is missing - this is where authors should list potential biases, limitations, comparison to previous studies, etc.

 Response 3: Thank you for the comment. Taking the advice, we have combined the discussion section to list potential biases and limitations with the conclusion section. Specifically, the revised paper includes the following discussions in page 12 line 341:

“The proposed signal control method in this study is developed for an intersection. A possible extension of this work is to factor signal coordination in a traffic network with multiple intersections. Traffic forecasts were not considered in the proposed method and additional research is needed to factor traffic forecast in the adaptive signal control methods. In addition, the signal phase in the proposed signal control method remains unchanged, another extension to this work is to study the method to adapt the signal phase along with signal timing based on detected traffic conditions.”

We also add some discussions related to limitations of the proposed method in the revised paper in page 5 line 161:

“The adapted signal timing is based on the detected conditions of the previous signal cycle and does not consider traffic forecasts. This is reasonable since that the traffic pattern within certain time period of a day is relatively consistent (similar to the basic premise of the non-adaptive traffic signal control methods), and the condition of the previous signal cycle is normally more reliable than traffic forecasts which tend to have high forecast errors.”

 Point 4: Throughout the paper, terminology is sometimes confusing. For example, section "Case study" (i.e. singular) mentions "several intersections" (which sounds like many), while in fact, two case studies were used. Authors are also inconsistent in using both "adaptive signal method" and "adapt signal method". They also unexpectedly mix past and present tense. In addition, the text contains high number of typographical errors (queen instead of queue, volumn instead of volume) and strange expressions ("years away", "signal timing is wasted", "polished this paper"). The mentioned points should be improved, including corrections by a native speaker.

 Response 4: Thank you for the comment. We have checked the English writing carefully, including problems related to terminologytypographical errorsword inconsistenttense etc. 

Round  2

Reviewer 1 Report

The paper partially respond to my revisions. In the new version of the paper, some of the new sentences have to be referenced. For instance, at line 56 you recall "big data and transport models approaches", that could be referenced with Birgillito et al. 2018 as I indicated in my previous revision. 

The numbers in the table 1 continue to not are clear. Which is the unit of measure? The day? the hour, ...?  Please specify it!

Author Response

Point 1: The paper partially respond to my revisions. In the new version of the paper, some of the new sentences have to be referenced. For instance, at line 56 you recall "big data and transport models approaches", that could be referenced with Birgillito et al. 2018 as I indicated in my previous revision.

 Response 1: Thank you for the comment. We have added the suggested reference.

 Point 2: The numbers in the table 1 continue to not are clear. Which is the unit of measure? The day? the hour, ...?  Please specify it!

Response 2: Thank you for the comment. To clarify Table 1, the revised paper includes the following statement in page 7 line 247:

“Table 1 shows the initial traffic volume of Linquan-Wenjing intersection in vehicle through per signal cycle length and the initial signal cycle length is 96 seconds. For example, in the third signal cycle, traffic flow of north straight is 5 which means that 5 vehicles travelled north straight during the third signal cycle. Note that the signal cycle length will change once the proposed adaptive signal control method is implemented.”

Reviewer 2 Report

Thank you for the revision. I can see that most of the comments were addressed, with exception of one. When I recommended to statistically compare the displayed differences, I meant using a 2-sample test (t-test or Kolmogorov-Smirnov test),in order to statistically verify the conclusions (Figures 8 and 9).Please include this into the study and form the conclusions accordingly.

Author Response

Point 1: Thank you for the revision. I can see that most of the comments were addressed, with exception of one. When I recommended to statistically compare the displayed differences, I meant using a 2-sample test (t-test or Kolmogorov-Smirnov test), in order to statistically verify the conclusions (Figures 8 and 9). Please include this into the study and form the conclusions accordingly.

 Response 1: Thank you for the comment. To statistically compare the displayed differences, we used matched-pairs t-test to test whether there is a significant mean difference between two sets of paired data. The results of matched-pairs t-test are added in line 297 of page 10:

“Matched-pairs t-tests are preformed to test whether there is a significant mean difference between total delays of each signal cycle before and after introducing the proposed method at both intersections. The results show that the total delays of each signal cycle of Linquan-Wenjing and Xinghu-Xiandai intersections are statistically significant (p=0.004 and p=0.009, respectively). It illustrates that the proposed method can lead to a statistically significant reduction in total delays at both intersections.”

Round  3

Reviewer 1 Report

The paper has been modified. 

Please, insert all units of measure in all Tables and Figures.

E.g. Table 1: Please, insert unit of measure in the caption 

Figure 8: Average delay is in "seconds"? 

Table 2: "Total delay of each signal cycle" are expressed in seconds?

Author Response

Point 1: The paper has been modified. Please, insert all units of measure in all Tables and Figures. E.g. Table 1: Please, insert unit of measure in the caption. Figure 8: Average delay is in "seconds"?  Table 2: "Total delay of each signal cycle" are expressed in seconds?.

 Response 1: Thank you for the comment. We have inserted the units of measure in all corresponding Tables and Figures in the revised paper.

Specifically, table captions are revised as follows: “Table 1. Traffic volume (vehicles/signal cycle) of Linquan-Wenjing intersection”; “Table 2. Traffic delay (seconds) reduction of Linquan-Wenjing and Xinghu-Xiandai intersections”; “Table 3. Traffic delay (seconds) under the adapted signal timing from different detector locations”.

For Figures, units of measure are inserted in the caption of the corresponding axis, i.e., Average Delay (seconds).

Reviewer 2 Report

Thank you for additional revision. Now I am happy to accept the paper and recommend it for publication.

Author Response

Thank you for the comment.